# Numerical Simulation Study on Deformation Characteristics of Surrounding Rock during Construction and Operation of Large Underground Gas Storage Structures

**Zhenhua Peng [1], Hao Ding [2,\*], Xinghong Jiang [2], Xuebing Hu [2] and Liang Cheng [2,\*]**

[1] CNOOC Petrochemical Engineering Co., Ltd., Qingdao 266104, China
[2] China Merchants Chongqing Communications Technology Research & Design Institute Co., Ltd., Chongqing 400067, China
\* Correspondence: dinghao@cmhk.com (H.D.); lightcheng@126.com (L.C.)

**Abstract:** Underground gas storage is an important technical measure for future natural gas storage. The stability of the surrounding rock during excavation and under ultra-high gas storage pressure is the key to the stable operation of gas storage reservoirs. A numerical calculation model for different surrounding rock conditions, different depth-span ratios, and different buried depth conditions was conducted to study the stability of surrounding rock after large section underground gas storage excavation and under an ultra-high gas storage pressure of 20 MPa. The results show that after construction is completed, the deformation of the rock surrounding the cavern increases with a decrease in the surrounding rock grade, and the deformation of the rock surrounding the cavern increases as the burial depth increases. In addition, the maximum vertical deformation of the surrounding rock decreases with the increase in the depth-span ratio of the cavern, and the maximum horizontal displacement increases with the increase in the depth-to-span ratio. While operating at 20 MPa gas storage pressure, the displacement of the rock surrounding the chamber tends to increase with the decrease in the surrounding rock grade and the deformation of the surrounding rock of the chamber decreases as the burial depth increases. Furthermore, the vertical displacement of the rock surrounding the chamber decreases with the increase in the depth-span ratio, while the horizontal displacement of the surrounding rock increases with the increase in the depth-span ratio. Considering the stability of the surrounding rock during construction and operation, gas storage chambers should be built in areas with better conditions, such as Grade II and Grade III surrounding rocks within a burial depth range of 200 m. Moreover, the stability of the surrounding rocks is better when the chamber depth-span ratio is 2.5~3.0. These research results can provide a theoretical reference for the design of large underground gas storage structures.

**Keywords:** underground gas storage; large section; surrounding rock stability; structural form; gas storage pressure; finite element

## 1. Introduction

China is a major energy consumer and the share of natural gas in China's total energy consumption is expected to reach approximately 15 percent by 2030. Natural gas reserves are an important aspect for guaranteeing the national energy supply and natural gas peak use, mainly using the three methods of: above ground tank storage, pipeline storage, and underground gas storage. Among them, underground gas storage includes the characteristics of a large gas storage capacity, saving ground resources, safety and reliability, low environmental pollution, and not being affected by weather, which is important for maintaining the normal use of natural gas and national strategic energy reserves [1,2]. With the support of successive national policies, the construction of underground gas storage will be further accelerated [3–5].

The cavern sections of underground gas storage facilities are large, and the ground stress is complex. Affected by cyclic gas injection and gas recovery, the stability of the surrounding rock is a key problem in the construction and operation of underground gas storage [6,7]. Much research has also been conducted by domestic and foreign scholars on the stability of rocks surrounding large-section caverns in underground oil and gas storage reservoirs. Lu et al. [8–10] calculated the radial displacement and plastic zone of the surrounding rock under a pressure of 25 MPa in the Swedish lined rock cavern storage reservoir using a three-dimensional finite element analysis method, and the obtained maximum displacement of 6 mm was consistent with the actual monitoring results. Zimmes et al. [11,12] used FLAC to calculate the plastic zone of the surrounding rock in circular chambers at different horizontal tectonic stresses, different internal pressures, and different chamber spacings, and devised reasonable spacings for the chamber arrangements. A. Suat Bagci et al. [13] analyzed a salt cave in central Turkey and obtained the optimal burial depth of 1275 m through computational analysis. It was found that the maximum chamber pressure increased with the increase in depth, while the minimum allowable chamber pressure also increased. Peng Zhenhua et al. [14,15] analyzed the stability of the surrounding rock of an underground water-sealed cavern reservoir, constructed in an island environment using a finite element numerical simulation method based on the flow-solid coupling theory. Yuan Weize et al. [16,17] studied the damage morphology and displacement deformation displacement law of the surrounding rock in large underground caverns during blasting. Zhang Chengbin et al. [18] studied a large underground cavern chamber using a FLAC3D numerical simulation and analyzed the stability of the cavern according to the surrounding rock stress and displacement. Hu Moupeng et al. [19] studied underground water-sealed oil storage reservoirs and analyzed cavern stability using a numerical simulation according to displacement criterion and stress criterion. Peng Jinghong et al. [20] successfully evaluated the stability level of four salt caverns containing multiple interlayers by establishing a comprehensive stability evaluation method. Yan Chunhe et al. [21–23] numerically simulated the time-yield deformation law and the extent of the time-yield damage zone at the top of the cavity in the surrounding rock and rock pillars of salt cavern gas storage reservoirs through the secondary development of ABAQUS. Pornkasem et al. [24] investigated the damage behavior of cavern envelopes under high internal pressure, as well as the mechanism of crack generation and development in the rock mass, by using a physical model test method on synthetic rock samples containing cavities. Xia Caichu et al. [25–27] addressed the stability of surrounding rocks in piezo gas storage in underground chambers operating under high internal pressure; the plastic zone and perimeter strain of the surrounding rocks in piezo gas storage chambers of different structural types and under high internal pressure was obtained through finite element calculations. At present, there is still insufficient research concerning surrounding rock conditions, the burial depth of cavities, and structure types in areas selected for the construction of high internal pressure underground gas storage. Therefore, it is important to carry out research on the deformation and plastic zone state of the surrounding rock of gas storage chambers during the construction and operation phrases for the future selection of underground gas storage sites.

In this study, a three-dimensional model of underground gas storage chambers is established using FLAC3D. The scale of a single gas storage reservoir, the characteristics of the deformation of the surrounding rock, and the change of the plastic zone of the large underground gas storage reservoir are analyzed under different surrounding rock conditions, different depth-span ratios and different burial depths after large section underground gas storage cavern excavation and under 20 MPa ultra-high gas storage pressure. From this analysis, the optimal structural type of the gas storage reservoir is obtained, which can provide theoretical support for the future design of large-section underground gas storage reservoirs.

## 2. Model

### 2.1. Cavern Structure Type

With a single vertical cave vault as the basic structure form, the effective volume of the cave chamber is designed according to $8.4 \times 10^4$ m$^3$, and the top of the chamber is hemispherical, the bottom is ellipsoidal, and the middle is connected by a cylinder (Figure 1). To obtain the optimal structural type of the cavern, it is necessary to study the influence of the structural depth-span ratio, surrounding rock grade, cavern burial depth and other factors on the stability of the surrounding rock during construction and operation, respectively.

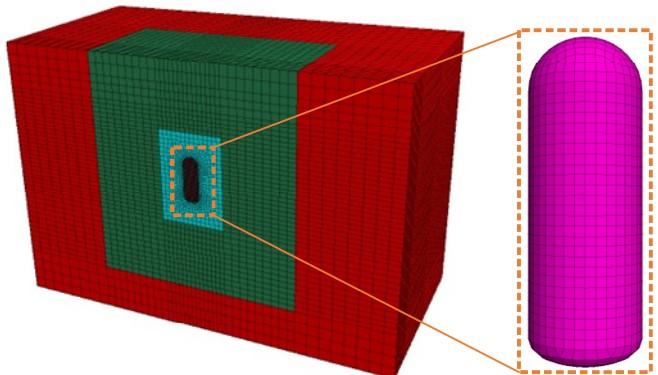

**Figure 1.** The finite element model of a single vertical cavern.

### 2.2. Constitutive Model

#### 2.2.1. Yield Criterion

When analyzing the stability of the surrounding rock of an underground gas storage reservoir, it is assumed that the surrounding rock material is an ideal elastic-plastic material. The yield criterion adopts the Moore-Coulomb criterion, and two damage modes of shear-$F_s$ and tension-$F_t$ are considered:

$$\left\{ \begin{array}{c} F_s = \sigma_1 - N_\varphi \sigma_3 + 2c\sqrt{N_\varphi} \\ F_t = \sigma_3 - \sigma_t \end{array} \right. \tag{1}$$

$\sigma_1$ is the maximum principal stress, MPa; $\sigma_3$ is the minimum principal stress, MPa; $c$ is cohesive force, MPa; $\varphi$ is the internal friction angle, ($^\circ$); and $\sigma_t$ is the tensile strength of rock, MPa.

#### 2.2.2. Plastic Flow Law

The plastic flow law of the surrounding rock material adopts the correlation flow law:

$$d\varepsilon_{ij}^p = d\lambda dg / \left( d\sigma_{ij} \right) \tag{2}$$

$\varepsilon_{ij}^p$ is plastic strain; $d\lambda$ is a non-negative plastic factor; $g$ is the plastic potential function for shear and tensile failure determined by the corresponding yield function; and $\sigma_{ij}$ is plastic stress, MPa.

#### 2.2.3. Unit Destruction

For calculation, the failure of the surrounding rock material adopts the second deviatoric strain invariant, the maximum tensile principal strain measurement, the plastic strain of rock shear and tension mode:

$$\left\{ \begin{array}{c} \varepsilon_s = \sqrt{\dfrac{\left(\varepsilon_1^p - \varepsilon_2^p\right)^2 + \left(\varepsilon_2^p - \varepsilon_3^p\right)^2 + \left(\varepsilon_3^p - \varepsilon_1^p\right)^2}{6}} \\ \varepsilon_t = \varepsilon_3^p \end{array} \right. \tag{3}$$

In the formula, $\varepsilon_s$ is shear strain; $\varepsilon_t$ is tensile strain; and $\varepsilon_1^p$, $\varepsilon_2^p$ and $\varepsilon_3^p$ are plastic strain corresponding to maximum principal stress, intermediate principal stress, and minimum principal stress, respectively.

When unit grid $\varepsilon_s \geq \varepsilon_{s,max}$ or $\varepsilon_t \geq \varepsilon_{t,max}$ is defined as the damaged unit grid.

*2.3. Boundary Conditions*

The horizontal boundary of the computational model is constrained by the displacement in the *x*-axis and *y*-axis, respectively; the lower boundary of the stratum is constrained by the displacement in the *z*-axis direction; the surface is a free boundary without any constraints. The actual buried depth above the top of the cavern is taken to calculate the surface deformation.

Since underground gas storage in China primarily occurs in salt cavern gas storage, and no high internal pressure tank gas storage facilities have been built yet, reference is made to a completed cavern gas storage project in Sweden. The corresponding working conditions are set to study the stability of the surrounding rock under different depth-span ratios, different surrounding rock grades and different burial depth conditions during the construction and operation phases (Table 1). The values of the parameters of the surrounding rock materials used in the calculation were taken based on the average values in GB/T 50218-2014 "Standard for Classification of Engineering Rock Masses" (Table 2) [23]. The calculation is divided into four steps. In the first step, considering the inherent weight of rock mass, the elastic solution method is used to generate the initial stress field. In the second step, the cavern excavation is simulated stage by stage according to the design process of the cavern type. In the third step, after each excavation, a sufficient calculation time step (20,000) is set to ensure that the surrounding rock stress of the cavern is fully realized along with the stress redistribution, and the displacement state of rock surrounding the cavern is also monitored. In the fourth step, after the excavation is completed, a uniform internal pressure of 20 MPa is applied to the inner wall of the cavern for subsequent calculation.

**Table 1.** Parameter settings for simulated working conditions.

| Working Condition | Depth-Span Ratio | Surrounding Rock Grade | Buried Depths/m |
|---|---|---|---|
| S1-1 | 1.5 | III | 200 |
| S1-2 | 2 | III | 200 |
| S1-3 | 2.5 | III | 200 |
| S1-4/S2-2/S3-2 | 3 | III | 200 |
| S1-5 | 3.5 | III | 200 |
| S2-1 | 3 | II | 200 |
| S2-3 | 3 | IV | 200 |
| S3-1 | 3 | III | 100 |
| S3-3 | 3 | III | 300 |
| S3-4 | 3 | III | 400 |
| S3-5 | 3 | III | 500 |

**Table 2.** Physical and mechanical parameters of the surrounding rock.

| Surrounding Rock Grade | Unit Weight /(kN·m$^{-3}$) | Elastic Modulus/GPa | Poisson's Ratio | Internal of Friction Angle/(°) | Cohesion/MPa | Calculation of Friction Angle /(°) |
|---|---|---|---|---|---|---|
| II | 26 | 27 | 0.2 | 55 | 1.8 | 75 |
| III | 24 | 13 | 0.28 | 45 | 1.1 | 65 |
| IV | 22 | 4 | 0.35 | 35 | 0.5 | 55 |

## 3. Results and Discussion

*3.1. Influence of Surrounding Rock Grade on Cavern Stability*

3.1.1. Deformation Characteristics of Cavern Surrounding Rock

As shown in Figures 2–4, under the conditions of 200 m burial depth and a fixed depth-span ratio of 2.5, after the excavation of the cavern is completed, the displacement of surrounding rock around the cavern increases with the decrease in the surrounding rock grade. With Grade II surrounding rock, the maximum settlement at the top of the cavern chamber during the construction phase is 0.8 mm, the maximum bulge at the bottom of the chamber is 2.6 mm, and the maximum horizontal displacement is about 2.9 mm. With Grade III surrounding rock, the maximum settlement at the top of the chamber is 3.6 mm, the maximum bulge at the bottom of the chamber is 17.7 mm, and the maximum horizontal displacement is about 9 mm. With Grade IV surrounding rock, the maximum settlement at the top of the chamber is 19 mm, the maximum uplift at the bottom of the cavity is 37 mm, and the maximum horizontal displacement is about 52 mm. As the height difference between the top and bottom of the cavern approaches 100 m, the resulting displacement at the top of the cavern is generally larger than the displacement at the bottom.

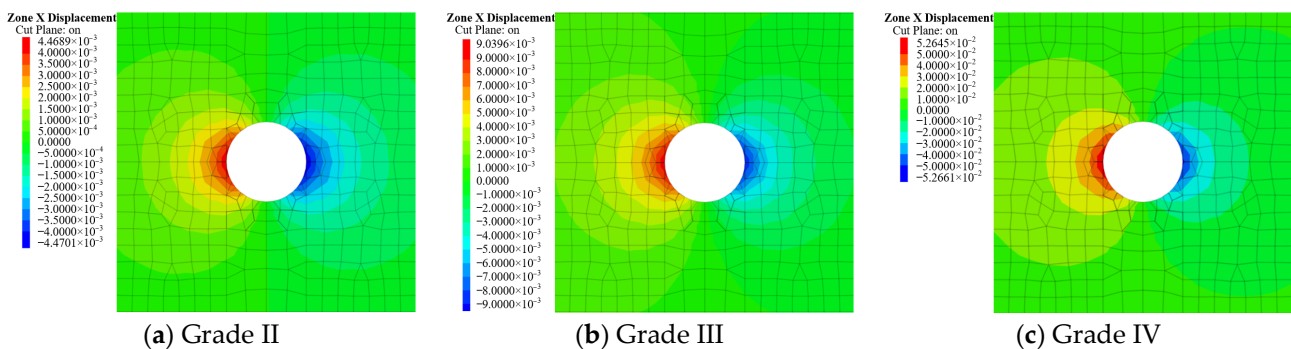

(**a**) Grade II　　　　　　(**b**) Grade III　　　　　　(**c**) Grade IV

**Figure 2.** Horizontal displacement nephogram of surrounding rock according to different rock grades during the construction phase.

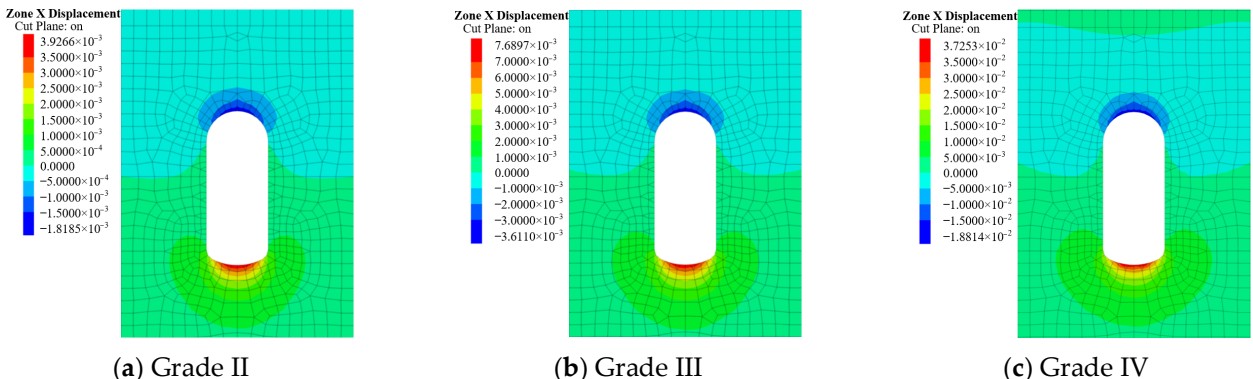

(**a**) Grade II　　　　　　(**b**) Grade III　　　　　　(**c**) Grade IV

**Figure 3.** Vertical displacement nephogram of surrounding rock according to different rock grades during the construction phase.

As shown in Figures 2–4, under the conditions of 200 m burial depth and 2.5 depth-span ratio, after applying equilibrium pressure of 20 MPa to the interior of the cavern, the displacement of surrounding rock around the cavern increases with the decrease in surrounding rock grade, showing an overall exponential growth trend. With Class II surrounding rock, the maximum displacements at the top of the cavern chamber during the operation period is 7.04 mm, the maximum displacement at the bottom of the cavern chamber is 7.05 mm, and the maximum horizontal displacement is about 12.8 mm. With

Class III surrounding rock, the maximum displacement at the top of the cavern chamber is 17.7 mm, the maximum displacement at the bottom of the cavern chamber is 15.4 mm, and the maximum horizontal displacement is about 29.6 mm. With Class IV surrounding rock, the maximum displacement at the top of the cavern chamber is 114.3 mm, the maximum displacement at the bottom of the cavern chamber is 68.3 mm, and the maximum horizontal displacement is about 148.1 mm. Due to the height difference of approximately 100 m between the top and bottom of the cavern chamber, the ground stress at the bottom of the cavern chamber is greater, which results in the displacements at the top of the cavern chamber being generally smaller than those at the bottom.

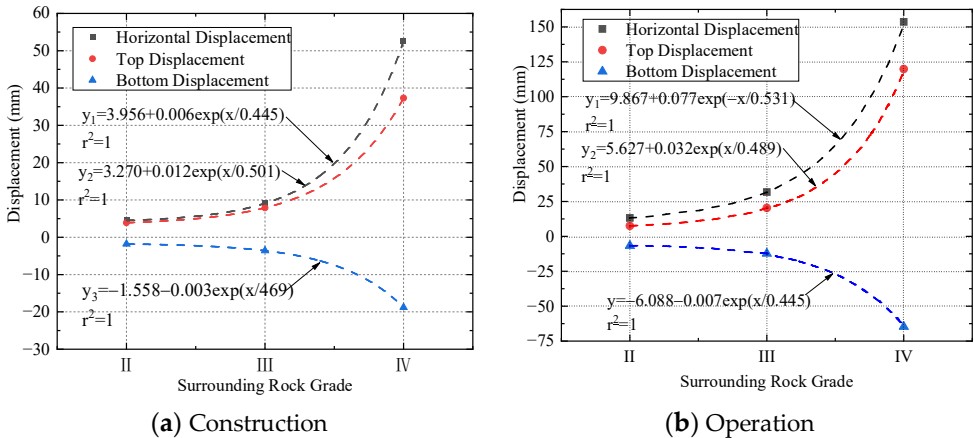

(**a**) Construction          (**b**) Operation

**Figure 4.** Maximum displacements of different surrounding rock conditions.

3.1.2. Distribution Characteristics of the Plastic Zone of Cavern Surrounding Rock

After the cavern is excavated, if the burial depth is 200 m and the depth-span ratio of 2.5 is fixed, the range of the plastic zone of the surrounding rocks around the cavern gradually increases as the grade of the surrounding rocks decreases (Figure 5). With Class II surrounding rock, the surrounding rock, as a whole, exhibits elastic deformation. With Class III surrounding rock, a small amount of plastic deformation appears around the bottom of the cavern (which may be due to the difference in height of about 100 m between the top of the cavern and the bottom of the cavern) and the difference in ground stress is large. However, with Class IV surrounding rock, plastic deformation is evident around the surrounding rock of the cavern, and the range of the plastic zone gradually extends to the interior of the surrounding rock.

Operation

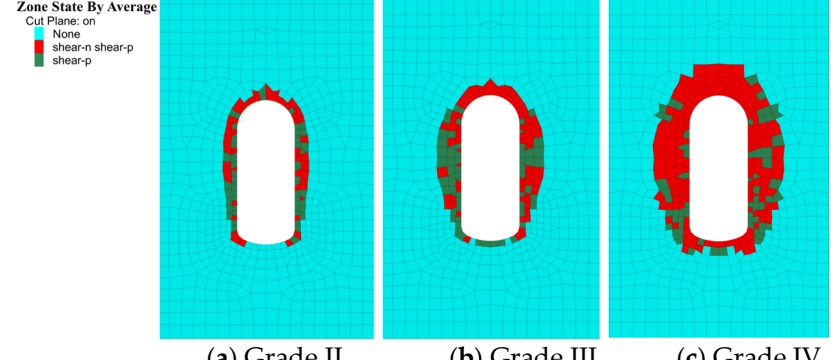

(**a**) Grade II          (**b**) Grade III          (**c**) Grade IV

**Figure 5.** Distribution of the plastic zone of surrounding rock according to different rock grades during the construction period.

If the burial depth is 200 m and the depth-span ratio of 2.5 is fixed, after applying equilibrium pressure of 20 MPa to the interior of the cavern, the range of the plastic zone of

the surrounding rock of the cavern increases gradually with the decrease in the surrounding rock grade (Figure 5). With Grade II surrounding rock, the entire surrounding rock exhibits elastic deformation. With Grade III surrounding rock, plastic deformation occurs around and at the bottom of the cavern. With Grade IV surrounding rock, plastic deformation is evident around the cavern, and the range of the plastic zone gradually extends to the interior of the surrounding rock. Due to the height difference of about 100 m between the top and the bottom of the cavern, the area and depth of the plastic zone of the surrounding rock at the top of the cavern are larger than the plastic zone of the surrounding rock at the bottom of the cavern.

According to these results, the cavern chamber should be arranged in areas with Grade II or better surrounding rock conditions. In areas with Grade III surrounding rock, a stronger lining is needed to ensure the stability of the overall structure. Areas with Grade IV or worse surrounding rock are not suitable for the construction of large underground gas storage chambers.

### 3.2. Effect of Depth-Span Ratio on Cavern Stability

3.2.1. Deformation Characteristics of Surrounding Rock

As shown in Figure 6, under the conditions of 200 m burial depth and fixed grade III surrounding rock, after the excavation of the cavern was completed, the settlement at the top of the surrounding rock and the rise at the bottom of the cavern showed an overall decreasing trend with the increase in the depth-span ratio. In addition, the maximum displacement in the horizontal direction of the cavern increased with the increase in the depth-span ratio. When the depth-span ratio is 2.5~3.0, the maximum settlement at the top of the chamber is about 3.5 mm, the maximum uplift at the bottom is about 7.5 mm, and the maximum horizontal displacement is 9 mm.

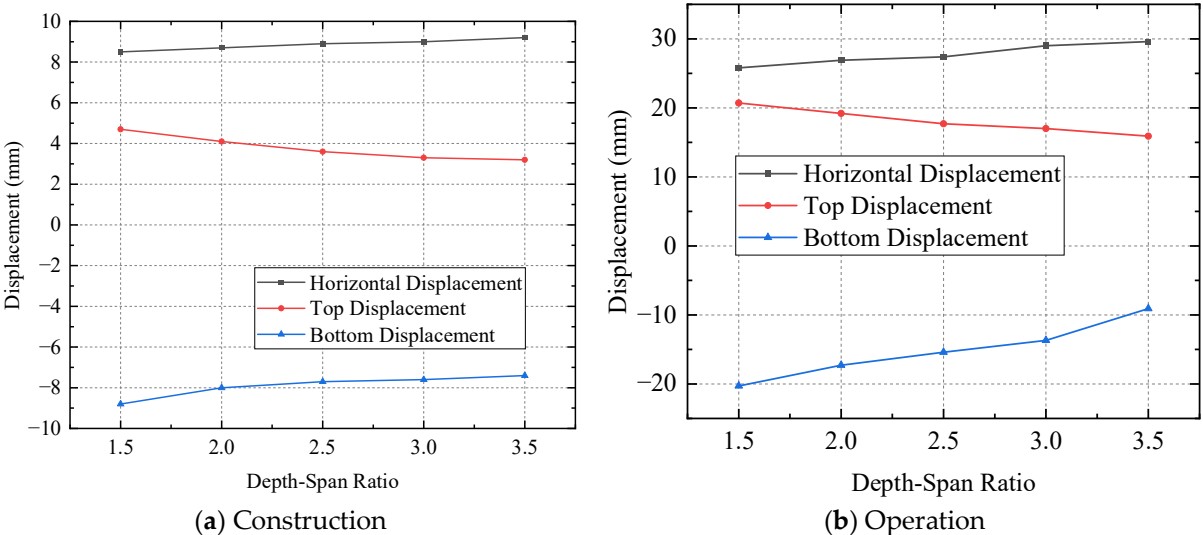

**Figure 6.** The maximum displacement variation diagram of surrounding rock under different depth-span ratios.

As shown in Figure 6, under the conditions of 200 m burial depth and Grade III surrounding rock, after applying equalization pressure of 20 MPa to the interior of the cavern, the maximum displacement of the surrounding rock in the horizontal direction of the cavern increases and then decreases as the depth-span ratio of the cavern increases, and the displacement of the surrounding rock at the top and bottom of the cavern decreases. When the depth-span ratio is 2.5, the maximum horizontal displacement is 29.6 mm, the maximum displacement of the top surrounding rock is 17.7 mm, and the maximum displacement of the bottom surrounding rock is 15.4 mm.

### 3.2.2. Distribution Characteristics of the Plastic Zone of Surrounding Rock

If the burial depth is 200 m and the Grade III surrounding rock are fixed, after the excavation of the cavern is completed, the area of the plastic zone of the surrounding rock gradually increases with the increase in the depth-span ratio, but the overall change of the plastic zone to the area ratio of the cavern structure is not significant. In addition, the plastic zone is mainly concentrated around and at the bottom of the cavern, and the area of the plastic zone at the bottom of the cavern increases with the increase in the depth-span ratio, mainly because the burial depth at the bottom of the cavern increases in relation to the increase in the depth-span ratio (Figure 7).

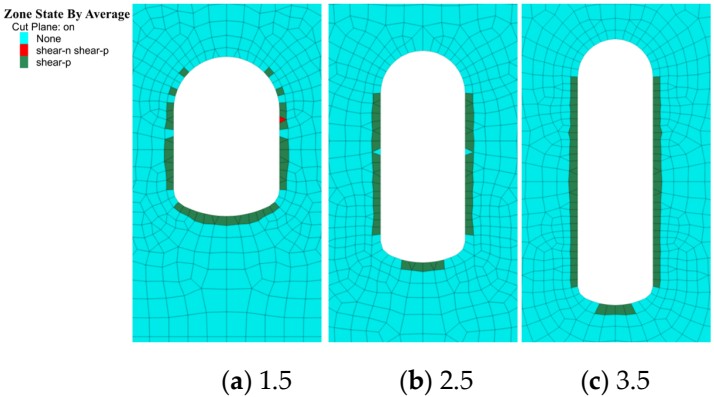

(**a**) 1.5 (**b**) 2.5 (**c**) 3.5

**Figure 7.** Distribution of the plastic zone of surrounding rock according to different depth-span ratios during the construction phase.

If the burial depth is 200 m and the Grade III surrounding rock are fixed, after applying equalization pressure of 20 MPa to the interior of the cavern, the area of the plastic zone of the surrounding rock gradually increases with the increase in the depth-span ratio. However, the overall change of the plastic zone to the area of the cavern structure ratio is not significant and is mainly concentrated around the cavern, as well as at the bottom of the cavern (Figure 8).

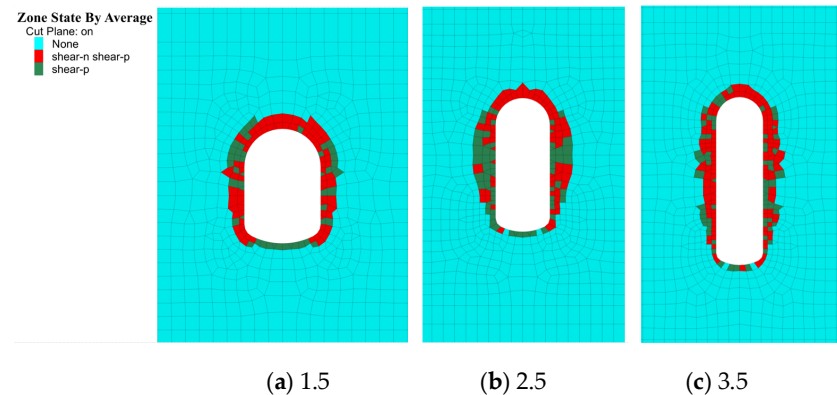

(**a**) 1.5 (**b**) 2.5 (**c**) 3.5

**Figure 8.** Distribution of the plastic zone of surrounding rock according to different depth-span ratios during the operation period.

According to the results, considering the influence on the stability of the surrounding rock during the construction and operation periods, the influence on the stability of the surrounding rock is minimized when the depth-span ratio of a large underground gas storage chamber is 2.5~3.0.

### 3.3. Effect of Burial Depth on Cavern Stability

3.3.1. Deformation Characteristics of Surrounding Rock

As shown in Figure 9, as the burial depth of the cavern increases, the settlement of the top of the surrounding rock and the horizontal displacement of the bottom bulge show an increasing trend, and the deformation of the surrounding rock at the mid-section of the cavern increases exponentially. The maximum settlement of the surrounding rock at the top of the cavern increased from 1.7 mm, at a burial depth of 100 m to 9.5 mm and at a burial depth of 500 m; the maximum uplift at the bottom of the cavern increased from 5 mm at a burial depth of 100 m to 17 mm at a burial depth of 500 m; the maximum horizontal displacement of the surrounding rock of the cavern increased from 5.1 mm at a burial depth of 100 m to 24 mm at a burial depth of 500 m.

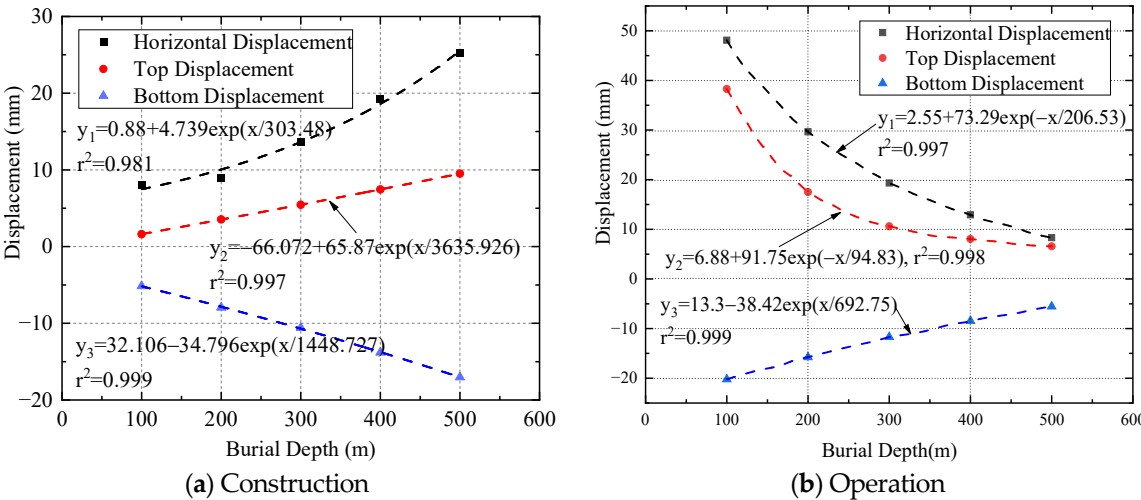

**Figure 9.** Maximum displacement variation of surrounding rock according to different burial depths.

As shown in Figure 9, with the increase in the burial depth, the top, bottom and horizontal displacement of the surrounding rock show an exponentially decreasing trend. The maximum displacement at the top of the chamber decreases from 38.8 mm at a burial depth of 100 m depth to 6.77 mm at a burial depth of 500 m; the maximum displacement of the bottom of the cave chamber decreases from 19.9 mm at a burial depth of 100 m to 5.34 mm at a burial depth of 500 m; the maximum horizontal displacement of the cave chamber decreases from 47.9 mm at a burial depth of 100 m to 8.51 mm at a burial depth of 500 m.

3.3.2. Distribution Characteristics of the Plastic Zone of Surrounding Rock

If there is Grade III surrounding rock and the depth-span ratio of 2.5 is fixed, after the completion of the excavation phase, the range of the plastic zone of the surrounding rock gradually increases with the increase in the burial depth of the chamber. At a burial depth of 100 m, the surrounding rock of the cavern chamber mainly deforms elastically, and plastic deformation begins to occur around it. Meanwhile, as the burial depth increases, the plastic zone of the rock surrounding the chamber gradually extends to the interior of the surrounding rock (Figure 10).

If there is Grade III surrounding rock and the depth-span ratio of 2.5 is fixed, after applying equilibrium pressure of 20 MPa to the interior of the cavern, the range of the plastic zone of the surrounding rock of the cavern gradually decreases as the burial depth of the cavern increases. At a burial depth of 100 m, the surrounding rock of the cavern results in damage to the plastic. At a burial depth of 500 m, the surrounding rock of the cavern gradually tends to display elastic deformation (Figure 11).

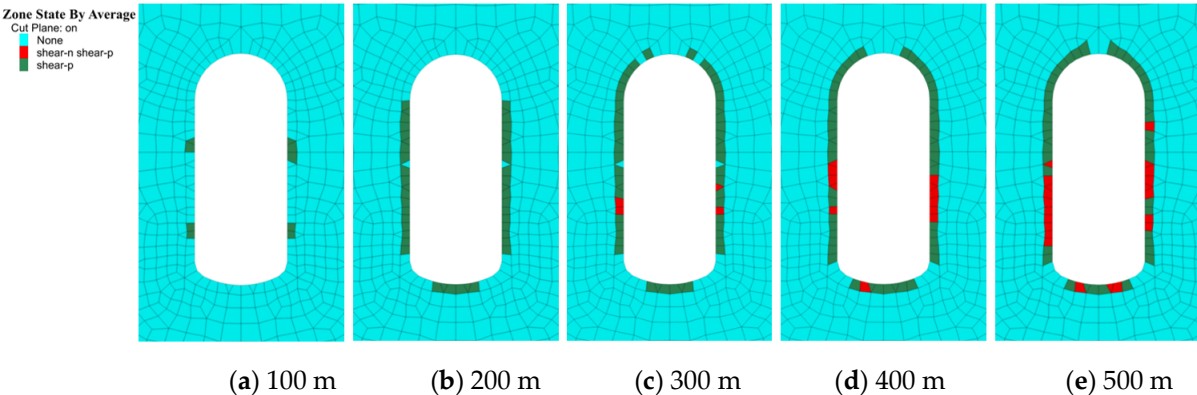

**Figure 10.** Distribution of the plastic zone of surrounding rock according to different burial depths during the construction phase.

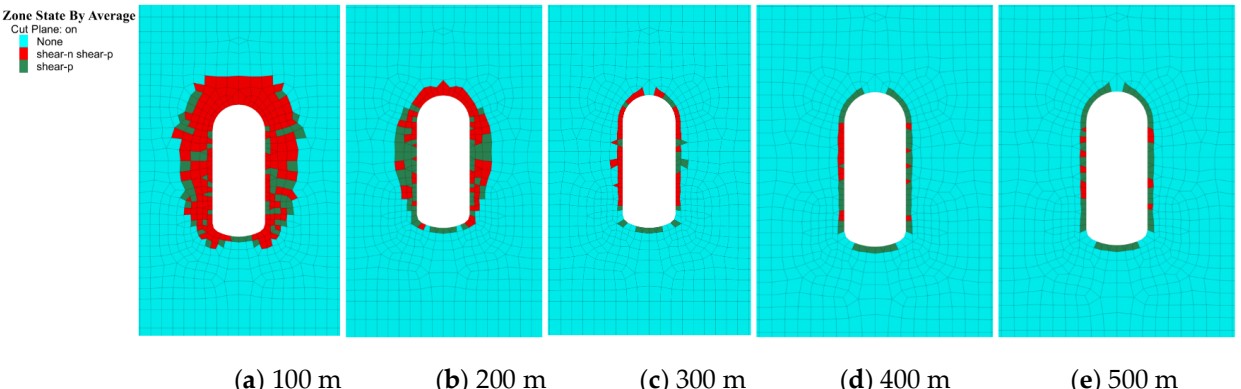

**Figure 11.** Distribution of the plastic zone of surrounding rock according to different burial depths during the operation period.

According to the results, to ensure the stability of the surrounding rock during the construction and operation phases, the burial depth of large underground gas storage chambers should be less than 200 m.

## 4. Conclusions

By conducting studies on the stability of the surrounding rock of underground gas storage chambers under different conditions (i.e., surrounding rock grades, depth-span ratios, and burial depths), monitoring the deformation characteristics of the surrounding rocks (under an ultimate gas storage pressure of 20 MPa applied inside the chambers during the operation period), and analyzing influences on the stability of the surrounding rocks of underground gas storage chambers, the following main conclusions were obtained.

1.  After construction, the deformation of the rock surrounding the cavern chamber increases with the decrease in the rock grade. During the operation period, under an ultimate gas storage pressure of 20 MPa, the displacement of the surrounding rock around the cavern chamber increases with the decrease in the surrounding rock grade. Therefore, the stability of Grade II and Grade III surrounding rock can, in general, be managed.
2.  After construction, the deformation of the surrounding rock around the cavern is less affected by changes in the depth-span ratio. During the operation period, if an ultimate gas storage pressure of 20 MPa is applied to the interior of the cavern, the vertical displacement of the surrounding rock in the cavern decreases with the increase in the depth-span ratio, while the horizontal displacement of the surrounding rock increases with the increase in the depth-span ratio.

3.  After construction, the deformation of the rock surrounding the cavern chamber increases as the burial depth increases. During the operation period, the displacement of the surrounding rocks around the cavern chamber, under an ultimate gas storage pressure of 20 MPa, decreases as the burial depth of the cavern chamber increases. This is mainly due to the increase in ground stress around the surrounding rock of the cavern chamber as the burial depth of the cavern chamber increases. However, the force on the surrounding rock will be smaller for the gas storage pressure inside the cavern.

4.  According to the results, considering the stability of the cavern chamber's surrounding rock during the construction period and operation period, the cavern chamber should be arranged in areas with good conditions, such as Grade II-III or better surrounding rock. Areas with Grade IV or worse surrounding rock conditions are not suitable for the construction of large underground gas storage chambers. The depth-span ratio should be 2.5~3.0, and the surrounding rock of the cavern chamber has better stability within a burial depth of 200 m.

**Author Contributions:** Methodology, Z.P. and H.D.; Formal analysis, X.H.; Investigation, H.D. and L.C.; Data curation, L.C.; Writing—original draft, X.J.; Writing—review & editing, L.C.; Funding acquisition, Z.P. All authors have read and agreed to the published version of the manuscript.

**Funding:** This research was financially supported by Chongqing Natural Science Foundation (Distinguished Youth Fund) project: cstc2021jcyj-jqX001.

**Institutional Review Board Statement:** Not applicable.

**Informed Consent Statement:** Not applicable.

**Data Availability Statement:** The datasets used and/or analyzed during the current study are available from the corresponding author on reasonable request.

**Conflicts of Interest:** The authors declare no conflict of interest.

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
