# Peer review of "Numerical Simulation Study on Deformation Characteristics of Surrounding Rock during Construction and Operation of Large Underground Gas Storage Structures"

_sustainability, doi:10.3390/su142416864_

Round 1

Reviewer 1 Report

The underground gas storage is an important guarantee to maintain the normal use of natural gas and national energy strategic reservesthe stability of the surrounding rock is the key problem so the study is of important values. A numerical simulation method was adopted and the deformation characteristics of surrounding rock was studied in this studyits content is rich. However, there are several questions

1、  Three points are not enough to obtain the overall exponential growth trend in Figure 4, and the parameters A1,y0, x was not calculated in the in the fitting equation, and the fitting accuracy need to be given.

2、  The stress of 20 MPa was choosed for the numerical simulation, and the reason need to explain.

3、  The pictures in figure 5,7,8,10 11 need more clearly.

4、  The serial No. should be 1)、(2)、(3and 4),not 1,2,3,4 in Conclusion part.

Author Response

Dear Editors and Reviewers

Thank you for your letter and for the reviewers’ comments concerning our manuscript entitled “Numerical Simulation Study on Deformation Characteristics of Surrounding Rock during Construction and Operation of Large Section Underground Gas Storage Structure” (ID: sustainability-2042484). Our great appreciation goes you for your valuable and constructive comments to improve our paper. We have revised the paper according to the comments. The responses to comments are attached as below.

Once again, many thanks for your suggestions.

Yours sincerely

Liang Cheng

Reviewer 1

Commen1: Three points are not enough to obtain the overall exponential growth trend in Figure 4, and the parameters A1, y0, x was not calculated in the in the fitting equation, and the fitting accuracy need to be given.

Response: Thank you for comments. Xia[1] and Xu[2] have been reported that the study of underground caverns is to select better surrounding rock conditions, so only grade 2, 3 and 4 surrounding rock are considered in numerical simulation. And the parameters A1, y0, x have been added Figure 4 and Figure 9 in the revised manuscript.

Reference:

[1] XU G C, YUAN W Z, XU J M, JIE X H, LI C X. Study on Excavate Scheme of Large Span Small Sagittal Ratio Underground Cavern [J]. Chinese Journal of Underground Space and Engineering, 2018, 14(S2):763-768.

[2] XIA C C, ZHANG P Y, ZHOU S W, ZHOU Y, WANG R. Stability and tangential strain analysis of large-scale compressed air energy storage cavern [J]. Rock and Soil Mechanics, 2014, 35(05):1391-1398.

Comment 2: The stress of 20 MPa was chosen for the numerical simulation, and the reason need to explain.

Response: A underground gas storage in rock cavern has been built in Sweden, the gas pressure is 15~25MPa during operation. And there are no other engineering cases, so 20MPa was chosen as the maximum operating pressure for numerical simulation. The References have been listed in the revised manuscript.

References:

[1] LU M. Finite element analysis of a pilot gas storage in rock cavern under high pressure [J]. Engineering Geology, 1998, 49(3-4): 353-361.

[2] JOHANSSON J. High-pressure storage of gas in lined rock caverns: cavern wall design principles [D]. Stockholm: Division of Soil & Rock Mechanics Royal Institute of Technology, 2003.

Comment 3: The pictures in figure 5,7,8,10 and11 need more clearly.

Response: All the pictures in the manuscript have been checked, and the blurred pictures have been improved in the revised manuscript.

Comment 4:The serial No. should be (1)、(2)、(3)and (4),not 1,2,3,4 in Conclusion part.

Response: Thanks for your suggestion. The serial No. have been revised in manuscript.

Reviewer 2 Report

A very good piece or research and numerical model, which is potentially useful for analysing the governing mechanisms for rock stability under high gas storage pressures. 

Author Response

Dear Editors and Reviewers

Thank you for your letter and for the reviewers’ comments concerning our manuscript entitled “Numerical Simulation Study on Deformation Characteristics of Surrounding Rock during Construction and Operation of Large Section Underground Gas Storage Structure” (ID: sustainability-2042484). Our great appreciation goes you for your valuable and constructive comments to improve our paper. We have revised the paper according to the comments. The responses to comments are attached as below.

Once again, many thanks for your suggestions.

Yours sincerely

Liang Cheng

Reviewer 2

A very good piece or research and numerical model, which is potentially useful for analyzing the governing mechanisms for rock stability under high gas storage pressures.

Response: Thank you for your recognition of our work.

Reviewer 3 Report

The authors have presented a numerical simulation study in this manuscript - however - it is extremely hard to follow what they are trying to say because of poor English. Sentence structuring is poor and grammatical mistakes are ubiquitous.

1) I highly recommend the authors rewrite the manuscript and consult a native English speaker while doing so take help from the English communication/literature department in their institution.

2) I suggest the authors to also provide experimental/field evidences that corroborates their numerical simulation work.

Author Response

Dear Editors and Reviewers

Thank you for your letter and for the reviewers’ comments concerning our manuscript entitled “Numerical Simulation Study on Deformation Characteristics of Surrounding Rock during Construction and Operation of Large Section Underground Gas Storage Structure” (ID: sustainability-2042484). Our great appreciation goes you for your valuable and constructive comments to improve our paper. We have revised the paper according to the comments. The responses to comments are attached as below.

Once again, many thanks for your suggestions.

Yours sincerely

Liang Cheng

Reviewer 3

The authors have presented a numerical simulation study in this manuscript - however - it is extremely hard to follow what they are trying to say because of poor English. Sentence structuring is poor and grammatical mistakes are ubiquitous.

Comment 1: I highly recommend the authors rewrite the manuscript and consult a native English speaker while doing so take help from the English communication/literature department in their institution.

Response: To improve the quality of the English grammar, we invited James Thompson from Chengdu Yestranslation Co., Ltd., China (http://www.yestrans.cn/) to edit the English text of this manuscript.

Comment 2: I suggest the authors to also provide experimental/field evidences that corroborates their numerical simulation work.

Response: Thanks for your suggestion. There only one engineering case around the world which the underground gas storage in rock cavern has been built in Sweden. And the purpose of this study is to obtain the key parameters of buried depth, depth-span ratio and surrounding rock conditions of underground gas storage structure through numerical calculation, and to provide preliminary support for engineering site selection and structural design in the application process of this technology. So this study failed to verify by actual cases. If there are project cases in the future, we will carry out further research according to the characteristics of the project.
